# Review of the JCAP/JATOP Air Quality Model Study in Japan

**Yoshiaki Shibata** [1,*] and **Tazuko Morikawa** [2]

1   Institute of Integrated Atmospheric Environment, Tokyo 112-0004, Japan
2   Japan Automobile Research Institute, Ibaraki 305-0822, Japan; tmorikaw@jari.or.jp
*   Correspondence: yshibata29130_faqri@qc.commufa.jp; Tel.: +81-3-6801-6082

**Abstract:** Around 1997, when JCAP (the Japan Clean Air Program) began, Japan's atmospheric environment did not meet the environmental standards for $NO_2$ and suspended particle matters (SPM), and strict reduction requirements for automobile exhaust gas were required. To achieve environmental standards, further cooperation between the automobile technology and fuel technology sectors was needed. In Europe and the United States, Auto-Oil programs were being implemented to reduce automobile exhaust gas, and JCAP was established as an Auto-Oil program in Japan. The Air Quality Model Study was one of the research themes and research activities continued for a total of 21 years, including JCAP I/II and JATOP I/II/III (the Japan AuTo Oil Program). JATOP was the successor program of JCAP. This paper describes the outline and main results of the JCAP/JATOP Air Quality Model Study.

**Keywords:** air quality; regional air quality model; roadside air quality model; emission inventory; micro-scale traffic model; secondary organic aerosol model; air quality observation

## 1. Introduction

There are general atmospheric measurement stations and roadside atmospheric measurement stations for atmospheric environmental concentrations in Japan. As shown in Figure 1, the atmospheric environment was gradually improved by measures against emission sources around 1997, but the attainment rate of $NO_2$/suspended particle matters (SPM, see Figure S1) was low at roadside measurement stations, which were greatly affected by automobile exhaust gas. This was one of the major issues for the atmospheric environment around the year 2000.

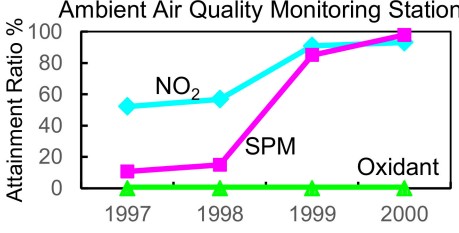
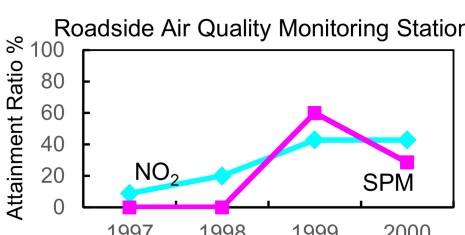

**Figure 1.** Attainment rate at general stations and roadside stations.

In order to improve Japanese air quality, further cooperation between the automobile technology and fuel technology sectors was needed. Since Auto-Oil programs were being implemented to reduce automobile exhaust gas in Europe and the United States, JCAP (the Japan Clean Air Program) was started as an Auto-Oil program subsidized by METI (the Japanese Ministry of Economy, Trade and Industry) in 1997. JCAP was in place until fiscal year 2006, and in 2007 it was succeeded by the research of JATOP (the Japan Auto Oil Program) which continued through 2017.

In both JCAP and JATOP, the Air Quality Model Study was one of the research themes. This paper describes the JCAP/JATOP Air Quality Model Study and the Automobile Emission Estimation Model developed for that purpose (Figure 2).

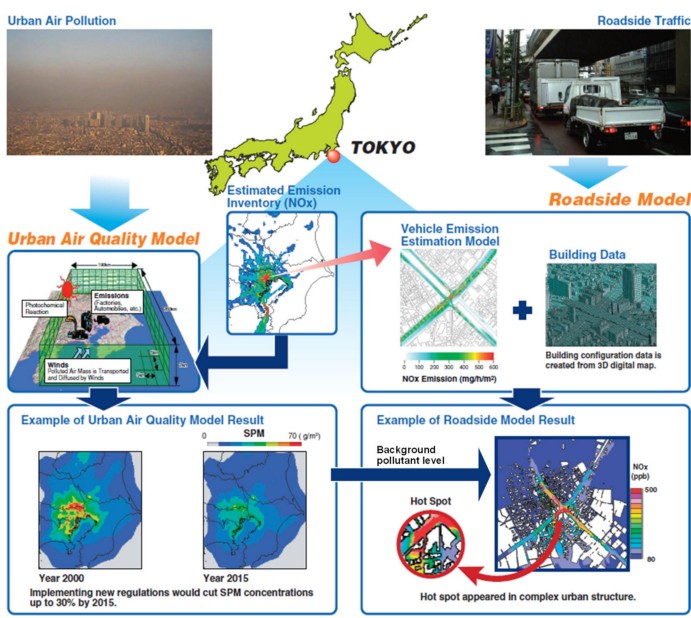

**Figure 2.** Framework of the JCAP/JATOP Air Quality Model Study.

The Air Quality Model Study was based on simulations, preparation of emission data and observation. Since simulation is indispensable for future estimation and sensitivity analysis, we have continued to conduct research in steps to improve the accuracy of simulations while comparing simulation results with observation results. To reproduce prediction values close to observed values, which we define "prediction accuracy", the emission inventory and the simulation model are still being improved. With a certain degree of accuracy secured, sensitivity analysis and future estimation will be conducted to examine the impact of automobiles and fuels on the atmospheric environment. The Air Quality Model Study was the first program in Japan to consistently carry out inventory establishment, Air Quality Model application, and future estimation/source analysis.

To achieve this framework, the following research was conducted:

1. Establish an emission inventory and improve accuracy;
2. Apply an Urban Air Quality Model to Japan and improve prediction accuracy;
3. Develop a Roadside Air Quality Model and improve prediction accuracy;
4. Predict the future air quality and analyze the direction of effective measures.

Through these activities, the obtained results were utilized "Contribution to Policy Making Processes" and "Emission Inventory Release".

As for Contribution to Policy Making Processes, the research results, such as verification results of effects of novel emission control technology on the atmospheric environment, were reported to an expert committee on motor vehicle exhaust emission: the Central Environment Council of the Ministry of the Environment. Through these activities, JCAP/JATOP Air Quality Model research has contributed to policy discussions by providing scientific knowledge.

As for Emission Inventory Release, the emission inventory developed by JCAP/JATOP Urban Air Quality Model research was released as JEI-DB. Additionally, JEI-DB became the basis for air pollutant emission inventory of the Ministry of the Environment. Emission Inventory Release has contributed to the progress of Japanese air quality modeling research.

Figure 3 shows an overview of the 21 years of the JCAP/JATOP Air Quality Model Study. To evaluate the air quality improvement effect of emission regulations, we started

a study in JCAP I to apply the chemical transport model, which has been popular in the United States, to Japan as an urban air quality model. Some characteristics of urban air pollution in Japan are that a large number of people are living near roads, severe traffic jams often occur at intersections of highways, buildings are lined up on both sides of roads to form street canyons and structures are complicated—such as elevated roads are covered. Roadside air quality was a major issue when the JCAP study was started in 1997, so we also conducted research on Roadside Air Quality Models that can more accurately evaluate the air quality near roads. The JCAP Roadside Air Quality Model was developed as a tool that enabled the study of scientific causal relationships by controlling vehicle emission, smoothing traffic, and changing geometric structures.

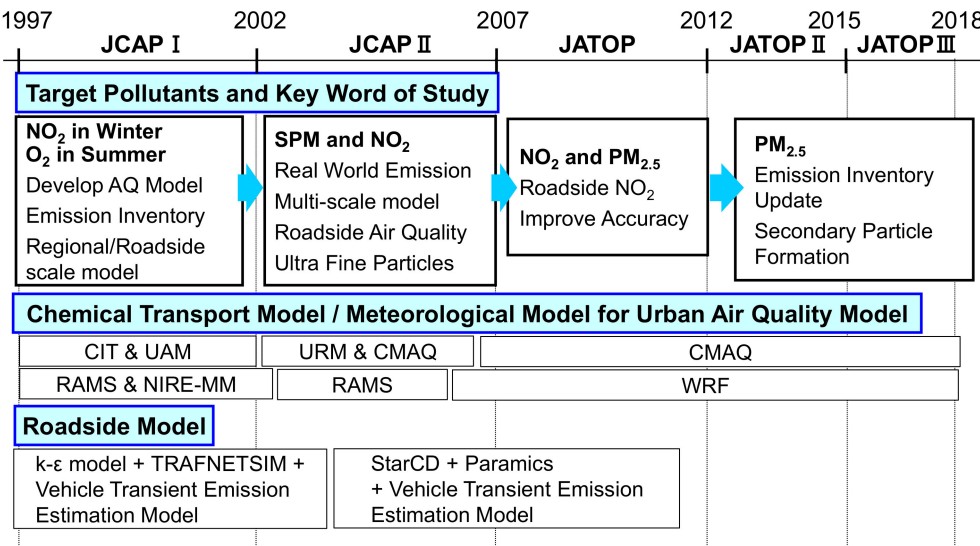

**Figure 3.** Overview of the 21 years of the JCAP/JATOP Air Quality Model Study.

The keywords of this study are establishment of emission inventory (EI) and application of an atmospheric environment model to Japan in JCAP I, improvement of EI and development of a multi-scale wide area model considering real world emissions in JCAP II, and the improvement of the prediction accuracy of roadside air quality and $PM_{2.5}$ in JATOP I. Additionally, the improved reproducibility of $PM_{2.5}$ and roadside $NO_2$, improved $PM_{2.5}$ prediction accuracy and improved EI accuracy after JATOP II, and improved SOA reproducibility in JATOP III.

During the JCAP Air Quality Model Study, the enhancement of emission inventory, high accuracy of the urban air quality model and development of a roadside air quality model were almost completed. In the JATOP Air Quality Model Study, the atmospheric environment model mainly was used for the future prediction of $PM_{2.5}$ and/or roadside $NO_2$ and analysis of the direction of effective measures. The scientific knowledge that was gained has contributed to national environment policies.

## 2. Urban Air Quality Model

### 2.1. Outline of Urban Air Quality Model

For the chemical transport model, CIT [1–3]/UAM [4,5] were applied for JCAP I, and both URM [6]/CMAQ [7] were applied to JCAP II for multi-scale support. After JCAP II, CMAQ was applied. The meteorological model initially applied was RAMS [8]/NIRE-MM [9] in JCAP I, but after JCAP II, WRF [10,11] was applied.

### 2.2. Emission Inventory for the Urban Air Quality Model

As for emission inventory, JCAP I developed an automobile emission model and created automobile emission data. JCAP II added an anthropogenic emission model other than automobiles and also natural sources emission such as biogenic VOCs and volcanoes.

Figure 4 shows the overall configuration of the emission inventory used in the JCAP/JATOP Urban Air Quality Model study. For anthropogenic sources, emission amounts are estimated separately, such as automobiles and fixed sources. For natural sources, biogenic VOC and volcano emission amounts, which affect PM and $NO_2$, are estimated. JEI-VEM (vehicles), G-BEAMS, biogenic VOC and volcano inventories will be described in the following sections.

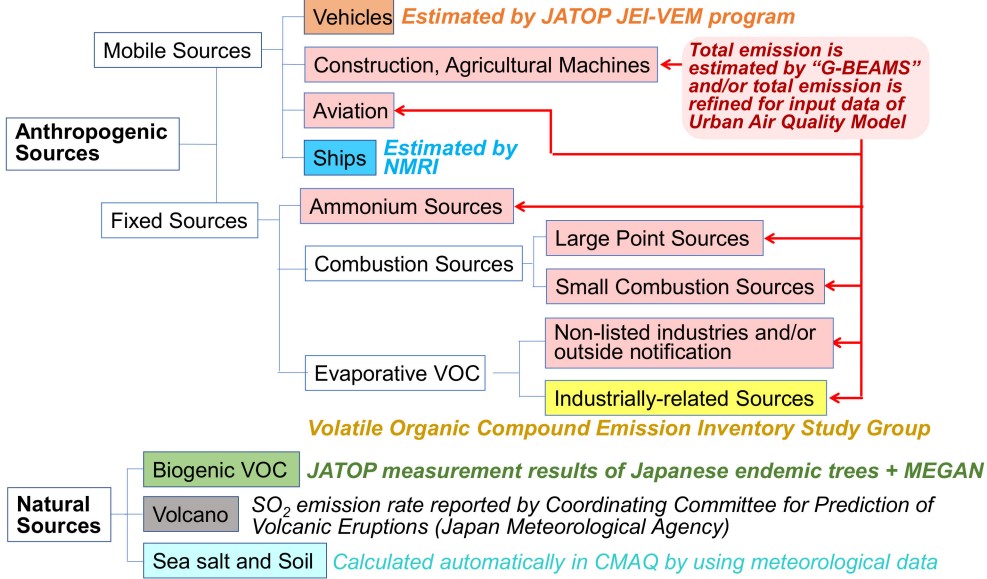

**Figure 4.** Overall configuration of the emission inventory for Urban Air Quality Model.

### 2.2.1. Automobile Emission

Outline of the Estimation Procedure

The JCAP/JATOP Emission Inventory for the Vehicle Emission Model (JEI-VEM) applies to tail pipe emissions during start-up and running, evaporation emission from gasoline vehicles, tire wear, and road dust (including brake dust). The tail pipe emissions are estimated separately for running and starting emissions. This is a bottom-up approach to estimate emission, although the concept of a top-down approach is also incorporated as the total mileage of vehicles which is matched to the mileage data of the national estimated value from the fuel consumption. Estimated pollutants are CO, NOx, $NO_2$, $SO_2$, THC, $CH_4$, PM, and $NH_3$. Figure 5 shows a conceptual diagram of JEI-VEM.

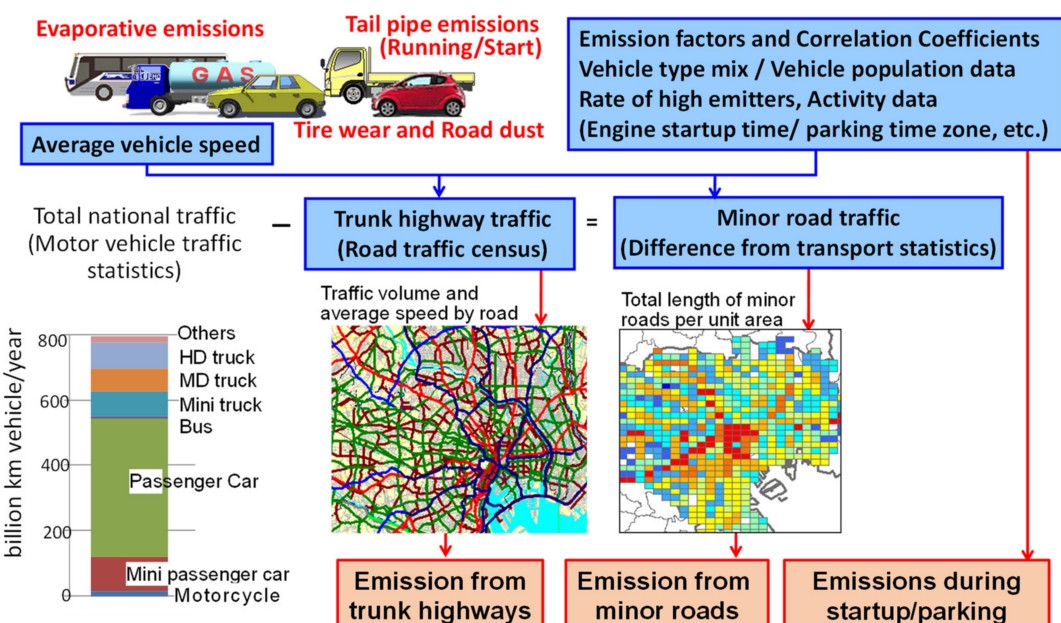

**Figure 5.** Conceptual diagram of JATOP Emission Inventory-Vehicle Emission Model (JEI-VEM), automobile emission estimation.

Emissions during driving are calculated by the product of the emission factor (EF) depending on vehicle speed for each emission regulation category and vehicle miles traveled (VMT). The EF is set for each regulation (Table S2). Since the vehicle category at the time of calculation reflects the usage of vehicles (business or private) and the vehicle weight of freight vehicles, 103 types of vehicle categories are set, including motorcycles (Table S3). For the VMT of main roads, traffic volume by simple vehicle category and average speeds per hour are based on the results of a road traffic census conducted every five years. The VMT of all roads is estimated from the Survey on Motor Vehicle Transport, and the minor road travel volume is estimated using the difference between total traffic volume and main road travel volume. The average vehicle speed on minor roads is uniformly set at 20 km/h.

Start and stop emissions are calculated by the product of EFs and the number of starts/stops. The numbers of starts/stops by vehicle category are estimated from the road traffic census OD survey, and the amount of activity of start/stop/parking for each mesh is assigned from the number of vehicles owned.

Evaporative emission is estimated by dividing it into DBL (Diurnal Breathing Loss, fuel evaporation gas generated while the vehicle is parked), HSL (Hot Soak Loss, fuel evaporation gas generated within 1 h after engine turn-off), and RL (Running Loss, fuel evaporation gas generated during running) according to the emitting process. RL is estimated from the EF, which is set from that measured in the exhaust gas test cycle, and VMT. Two types of EF are set before and after the introduction of evaporation regulation. HSL calculates a uniform EF over the number of stops regardless of vehicle type, regulations, and temperature. DBL is expressed by a function that depends on temperature change, canister capacity, and tank capacity [12].

For tire wear and road dust, EF is set for passenger cars and trucks, respectively, and it is calculated by applying VMT [13].

To reproduce real world emissions, various correction factors are applied to the emissions obtained from EF and VMT. To correct the influence from environmental conditions, a temperature correction coefficient is applied. For NOx, a humidity correction factor is also applied. A deterioration correction coefficient is applied to the performance deterioration of the aftertreatment device according to the vehicle age. For diesel vehicles with a diesel particulate filter (DPF), the frequency of DPF regeneration and the increase in emissions during DPF regeneration are reflected as the DFP regeneration deterioration factor. In

addition, if the engine is started within 12 h after the engine stopped, the emission amounts at the time of starting are corrected by the soak time correction coefficient.

All emissions are estimated spatially as 30″ in latitude and 45″ in longitude, equivalent to an approximately $1 \times 1$ km resolution (Figure 6), temporally as hourly by month.

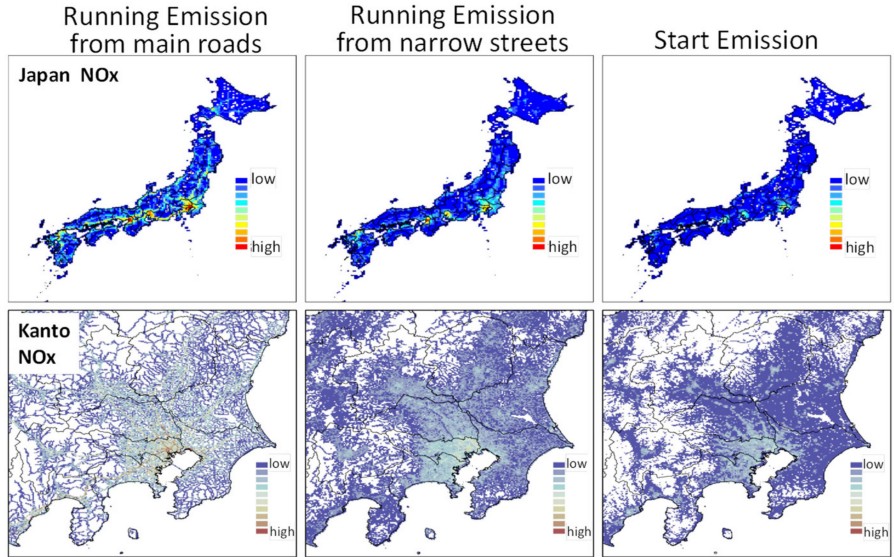

**Figure 6.** Example of automobile emission results by the JATOP Emission Inventory-Vehicle Emission Model (JEI-VEM).

High Emitting Vehicles

One of the notable studies of JCAP/JATOP was the study of high emitters. In estimating real-world emissions, various studies have pointed out that a certain percentage of cars in real in-use driving emit more emissions than the regulated level of emission due to breakdowns and poor maintenance. Since the existence of high-emitting vehicles was not recognized in Japan in the early 2000s, JCAP measured high-emitting vehicles by remote sensing device (RSD) measurement.

RSD is a device that measures the emissions of vehicles traveling on the road in real time with the configuration shown in S4. JCAP/JATOP conducted RSD measurements 6 times and surveyed nearly 350,000 vehicles (valid data 189,275, see Table S4).

The results are shown in Figure 7. The figure shows the frequency distribution of NO concentration by first registration year of passenger cars in the 1978 regulation category. The concentration distribution width increases as the number of years of driving increases. Since there is no clear standard for which concentration should be used as a high emitter determination, we used the cut point of the US in-use vehicles and calculated the frequency of vehicles with a higher concentration and use it in JEI-VEM as the high emitter vehicle ratio. However, since the year 2000 regulation (NST) of vehicles, exhaust gas regulations have become stricter, and the ratio of high emitter vehicles has dropped significantly.

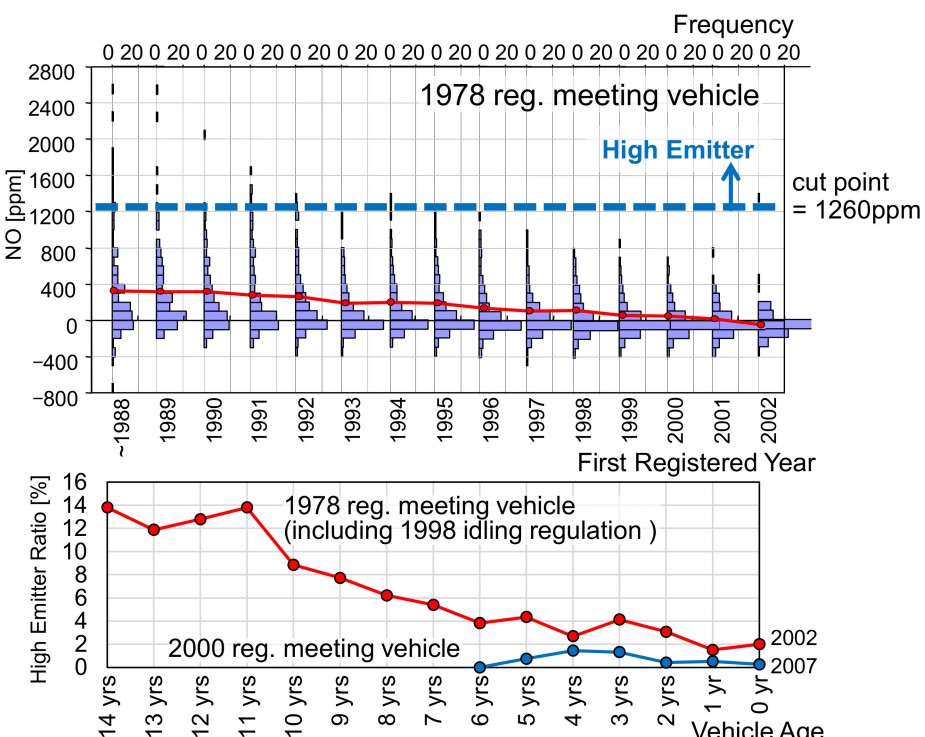

**Figure 7.** RSD measurement results and high emitter ratio of passenger cars.

Most of the high emitter vehicles are gasoline vehicles with a three-way catalyst, so we measured the EF when the catalyst failed (=without catalyst) [14].

### 2.2.2. Emission Inventory for Other Sources

G-BEAMS: Georeference-Based Emission Activity Modeling System

For emission from anthropogenic sources other than automobiles, the Georeference-Based Emission Activity Modeling System (G-BEAMS) has been developed in collaboration with the National Institute of Environmental Science. Figure 8 shows the overall outline of G-BEAMS. This is a top-down inventory that allocates nationwide emissions and is calculated by the EF and the activity data from sources such as domestic energy statistics and various industry statistics and emission data. Those emissions are then allocated by geographical and temporal resolution, using the operating hours of the source, etc. The total amount of evaporated VOCs estimated by the Ministry of Environment Japan (MOEJ) based on industry data on solvents, paints, adhesives, etc. [15], is spatiotemporally distributed by G-BEAMS.

In both cases, the spatial resolution is approximately $1 \times 1$ km to $10 \times 10$ km resolution, and the time resolution is 1 h per month.

### Biogenic VOC Sources

Biogenic VOCs have independently been modeled with trees unique to Japan, and other than that, MEGAN (Model of Emissions of Gases and Aerosols from Nature) [16,17] is applied. Figure 9 shows an overview of Biogenic VOC inventory. The basal EF is calculated by measuring the VOC emissions from the top eight species of broad-leaved and coniferous trees unique to Japan using the branch enclosure method [18]. The target components are isoprene, monoterpenes, and sesquiterpenes [19].

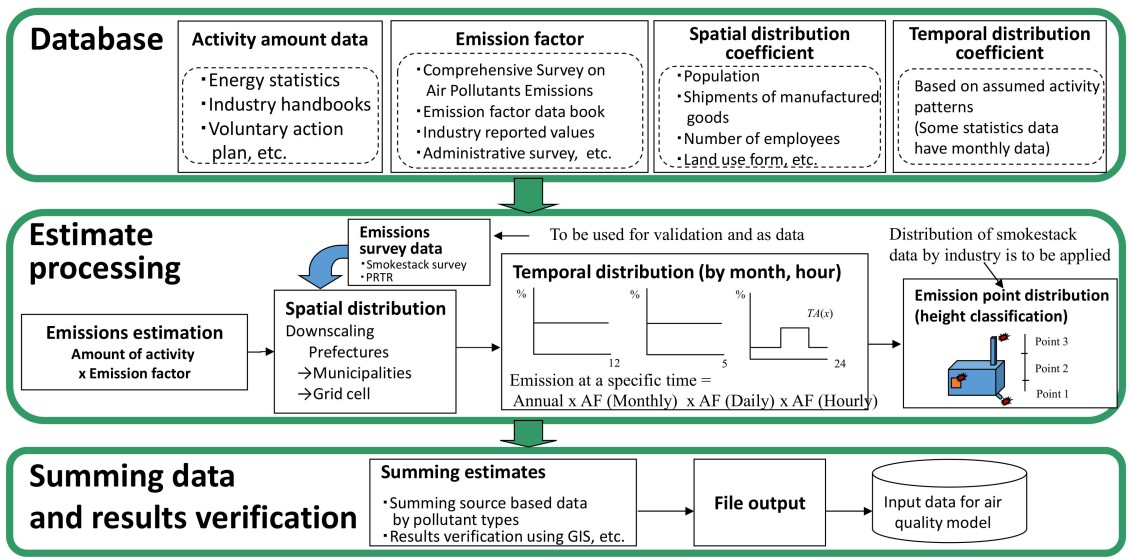

**Figure 8.** Outline of G-BEAMS.

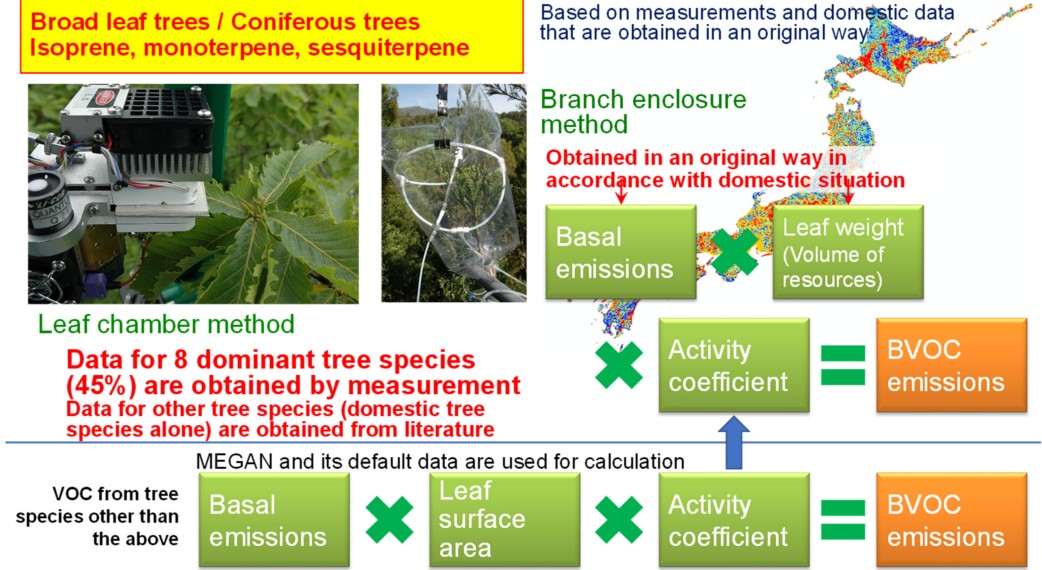

**Figure 9.** Overview of biogenic VOC inventory in JATOP.

VOC emissions are estimated from leaf weight and activity rates such as temperature and sunshine duration from basal release and national vegetation distribution data [20].

Volcanic Emissions

There are some global databases of SO$_2$ emissions from volcanoes, but not all active volcanoes in Japan are covered [21,22]. Therefore, JATOP has created a database of emissions from volcanoes from the materials of the Japan Meteorological Agency Volcano Eruption Prediction Liaison Committee. Figure 10 shows SO$_2$ emissions from 15 active volcanoes nationwide in 2015.

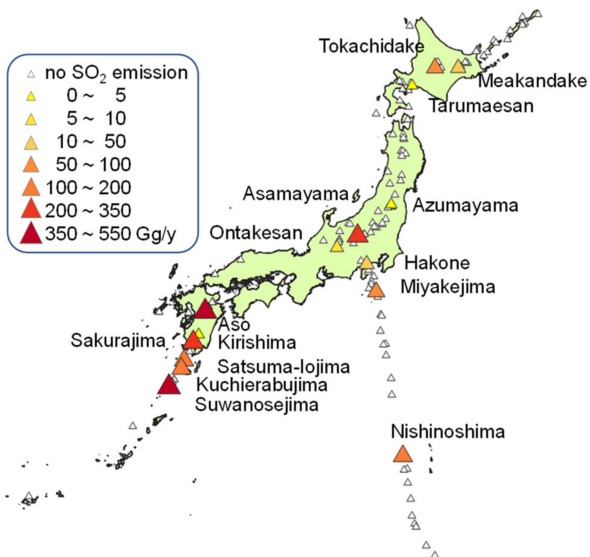

**Figure 10.** Active Japanese volcanos [23] and SO$_2$ emissions from volcanoes (2015).

In Japan, SO$_2$ emissions from volcanoes in 2015 were 1811 Gg, which is 5.4 times higher than anthropogenic SO$_2$ emissions, excluding ships.

Finally, for ship emissions, we applied ship data prepared by NMRI (the National Maritime Research Institute) based on ship traffic on domestic and external routes (Figure S5).

### 2.3. Urban Air Quality Model

2.3.1. Outline

The calculation area of the JCAP/JATOP Urban Air Quality Model has expanded from the Kanto area in JCAP I to the Japan area/Kanto area in JCAP II, and the East Asia area/Japan area Kanto area in JATOP.

As shown in Figure 11 in the JCAP II Urban Air Quality Model, a chemical transfer model is used for each region from the East Asian region. CMAQ is used for wider areas, and URM is used for the Kanto area. As a result, the URM model can nest the grid size from a 4 km mesh to a 1 km mesh in the Kanto area, which is the Tokyo metropolitan area. RAMS is used for the meteorological model, and the simple urban canopy model (SUMM) [24] is introduced for RAMS in the area corresponding to the smallest grid of URM.

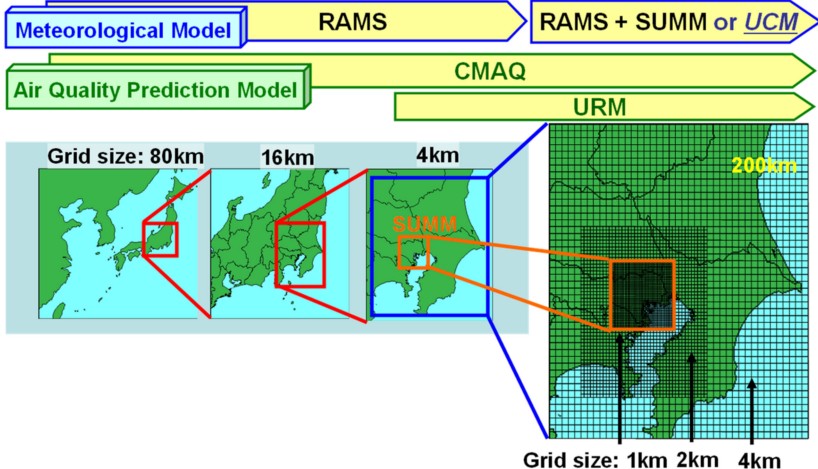

**Figure 11.** Configuration of the JCAP II Urban Air Quality Model.

In the JATOP Urban Air Quality Model, a 4 or 5 km mesh size is applied to the Kanto area due to the balance between calculation time and prediction accuracy, therefore, the CMAQ model can be used for the East Asian region through the Kanto area. The meteorological model also uses WRF, which can be combined with CMAQ. Figure 12 shows the JATOP Air Quality Model specifications [25].

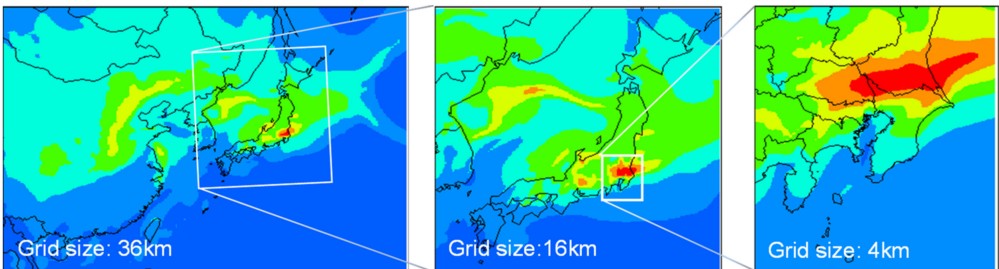

**Figure 12.** Configuration of JATOP Urban Air Quality Model.

### 2.3.2. Secondary Organic Aerosol (SOA) Model

Particulate matter is divided into two classes: primary generation and secondary formation. A more complex model is necessary for secondary formation than for primary generation because not only quantification of precursor gas emissions but also a reaction mechanism is needed for a secondary formation model. For secondary aerosol formation, the reaction between secondary organic aerosol (SOA) and secondary inorganic aerosol is taken into consideration. In the early 2000s, there was a lack of knowledge about SOA generation, but JCAP/JATOP has made improvements to the SOA generation model using chamber experiments (Figure S6-1,2) [26].

Based on the results of a photochemical chamber experiment, the SOA formation model included in the air quality model (CMAQv4.7.1) was modified and improved as below.

1.  The reaction parameters when secondary organic compounds (SOA) are generated from plant-derived VOCs (only NO non-coexistence reactions specific to forests are extracted) were updated.
2.  A further pathway was added for the reaction product to react with OH.

S6 shows the detail of these improvements.

### *2.4. Application of Urban Air Quality Model*

### 2.4.1. Future Air Quality Prediction

Future Emission Inventory Scenario

Based on the 2010 inventory, the future emission inventory for the fiscal year 2025 is estimated by changing the activity amount to the future forecast value. For the amount of activity, we used the basic plans of public institutions such as those of the national government. If the index value of the future estimation for 2025 could be utilized, we used those values, and if it could not be obtained directly, we estimated it by linear interpolation, etc. (example: interpolation of the 2020 index value and 2030 index value).

For vehicle emission inventory, the traffic volume/number of vehicles owned were used as the future forecast value, and the alternative conditions for vehicles were estimated in two cases (the case where substitution progresses and the case where substitution does not progress).

Figure 13 shows future emission inventory for 2025. All emissions are reduced by 40~5% from 2010FY to 2025FY.

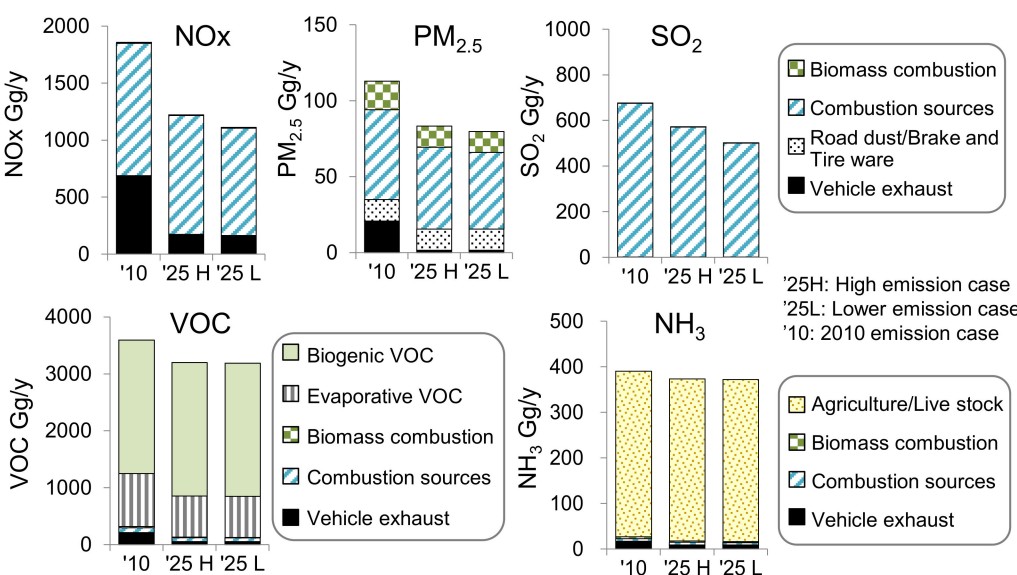

**Figure 13.** Future emission inventory for the 2025 fiscal year.

In JCAP/JATOP Air Quality Model research, inventory for fiscal years 2000/2005/2010 was also created. The future forecast is 2020/2025 in JCAP/JATOP Air Quality Model research.

Future Air Quality Prediction

Figure 14 shows the prediction results of future air quality in 2025. The reproducibility of the model for PM$_{2.5}$ with its components and O$_3$ in 2010 and 2012 are shown in Figure S7-1,2.

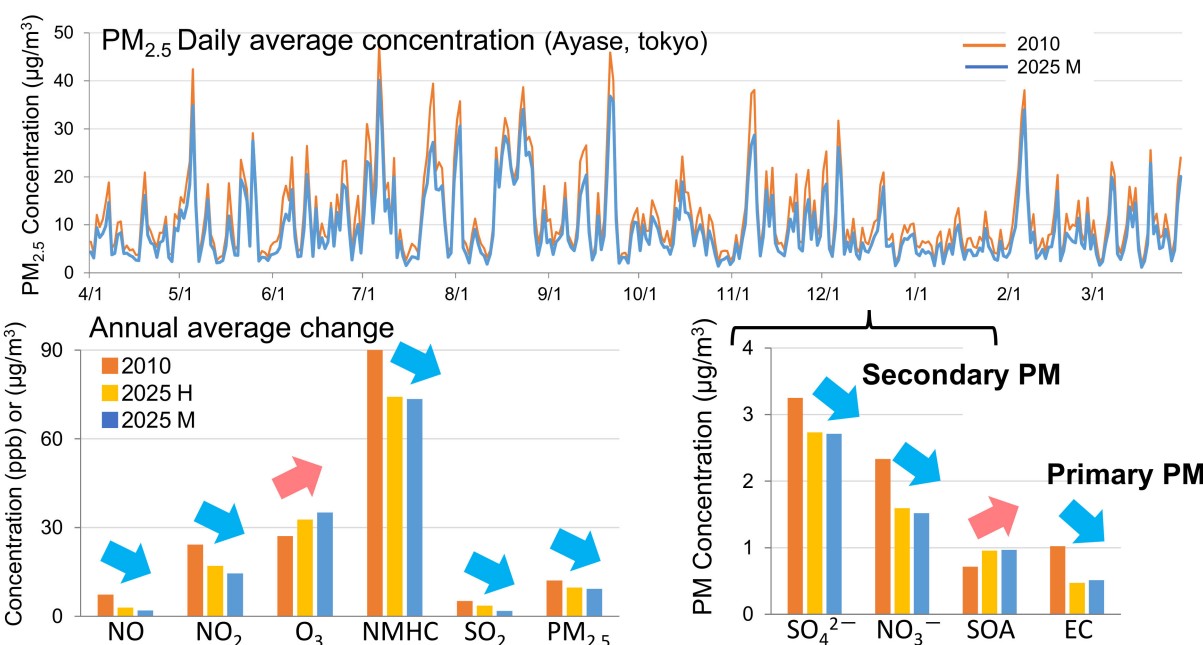

**Figure 14.** Future air quality prediction results for fiscal year 2025.

The concentrations of NOx, SO$_2$, and NMHC, which are PM$_{2.5}$ precursors, are lower, and the concentration of PM$_{2.5}$ is also slightly lower. On the other hand, the O$_3$/SOA concentration of the secondary substances is higher, but the sulfate nitrate is lower.

### 2.4.2. Source Sensitivity Analysis

For study of the JATOP Air Quality Model, to evaluate the effect of countermeasures for each source, the non-linearity of $PM_{2.5}$ accompanied by reaction generation in the atmosphere was considered, and as a realistic condition, the emission reduction of 20% was selected for sensitivity analysis (Figure 15) [27].

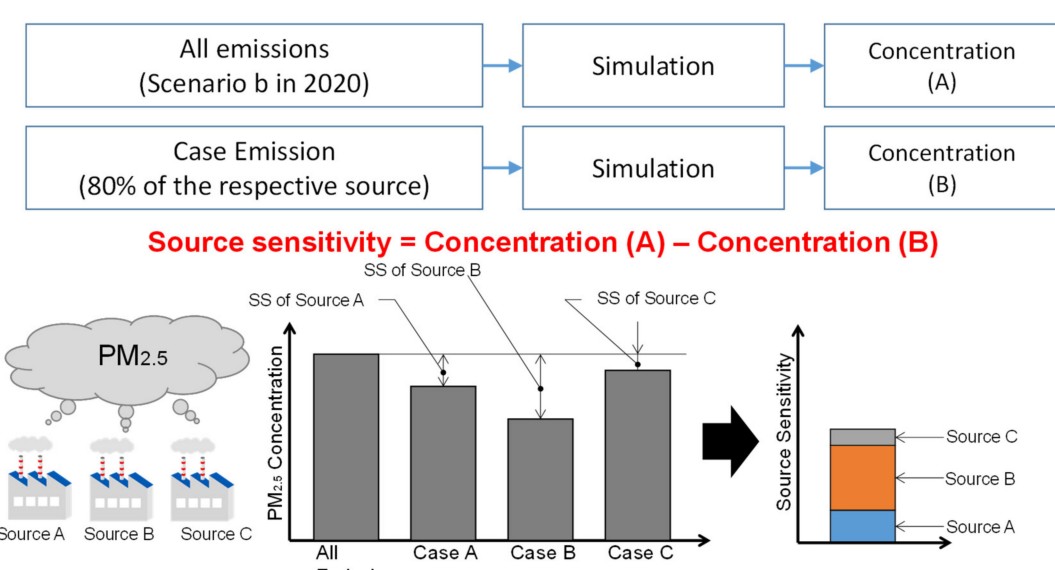

**Figure 15.** JATOP source sensitivity analyses.

The results are shown in Figure 16. The Tokyo metropolitan area was analyzed.

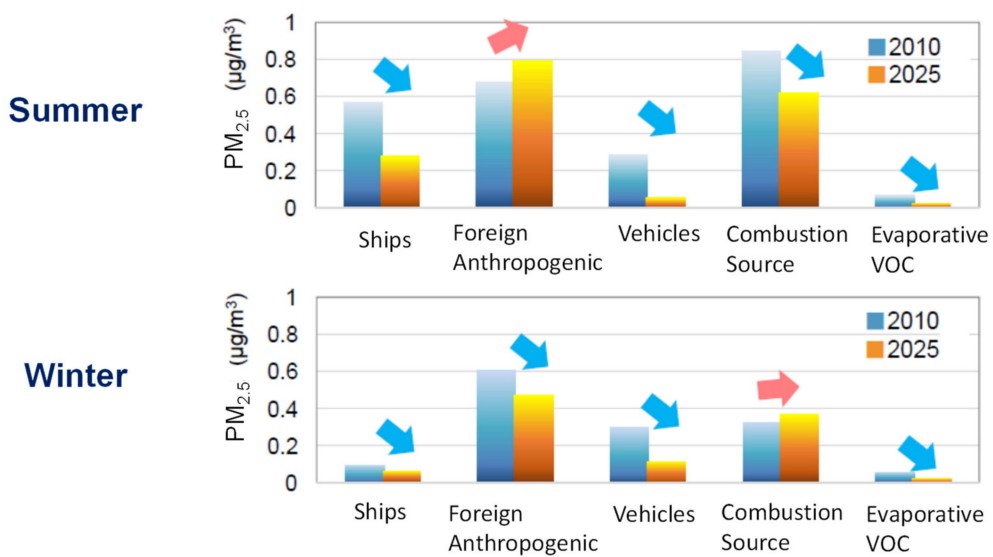

**Figure 16.** Source sensitivity analysis results (Tokyo).

It was shown that the sensitivity of ships and automobile sources for which emissions are decreasing due to regulations, etc., will decrease, but the sensitivity of inflows from abroad will increase.

## 3. Roadside Air Quality Model

### 3.1. Outline

The JCAP Roadside Air Quality model consists of simulation of micro-scale traffic on target roads, estimation of emission distribution along the roads, and simulation of air flow and advection/diffusion of emission around the roads. Roadside air concentration is considered by the sum of background air concentration and direct contribution of automobile emission on roadsides. The background concentration is the grid average concentration of the Urban Air Quality Model.

The scheme figure of the Roadside Air Quality Model is shown in Figure 17. For JCAP I, the κ-ε model is initially applied to the fluid model [28], TRAFNETISM [29] is applied to the traffic flow model, and the transient vehicle emission estimation model was developed. After JCAP II, Star-CD® (Siemens AG, Munich, Germany) applied to the fluid model, and Paramics® (SYSTRA Ltd, Edinburgh, UK) is applied to the traffic flow model. The background concentration is the grid average concentration of the Urban Air Quality Model. The model is verified by comparison with the wind tunnel experiments and tracer gas concentration in a diffusion field experiment in a real urban area and pollutants level at roadside air monitoring stations [30,31].

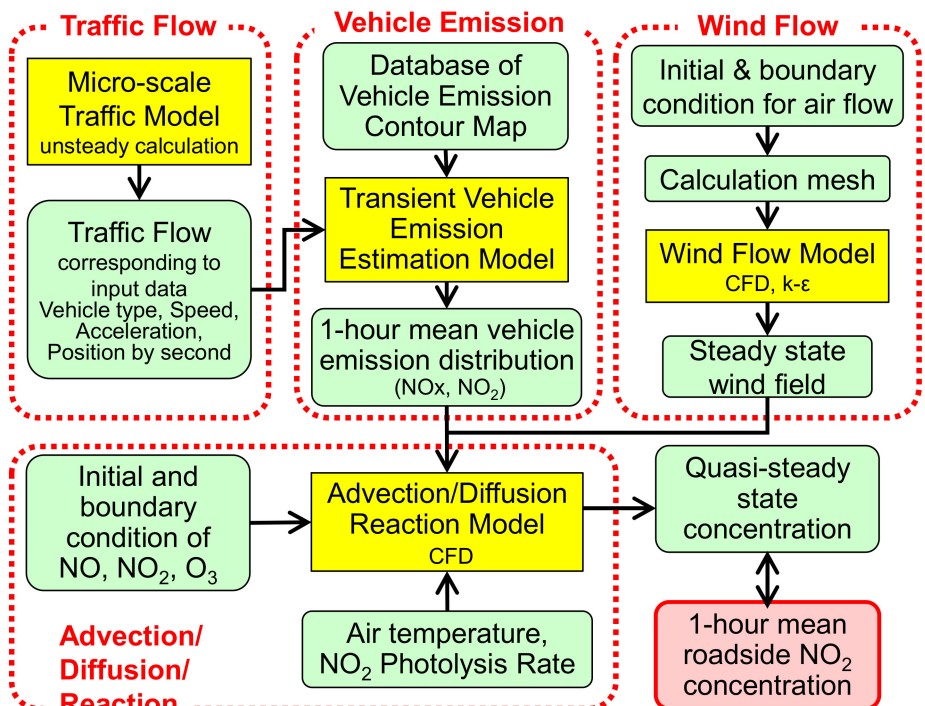

**Figure 17.** Structure of the Roadside Air Quality Model.

The roadside model predicts five locations in the JCAP/JATOP Model Study (Kamiuma, Yamato, Ikegami, Noge, Sengenshita, see Figure S11-2).

### 3.2. Transient Emission Inventory Model

The Transient Emission Inventory Model (Figure S8-1–3) consists of a combination of the "emission map" created by a chassis dynamometer test and the "micro-scale traffic model". More than 50 types of emission maps were created in the three years of JCAP II. Individual vehicle position, speed, and acceleration were estimated every second by the micro-scale traffic model (Figure S9).

These results can be integrated over time to obtain an hourly average emission distribution from automobiles. Figure 18 shows the distribution of NOx and PM emissions at the one of the most polluted intersections in Japan: Kamiuma (Setagaya-ku, Tokyo).

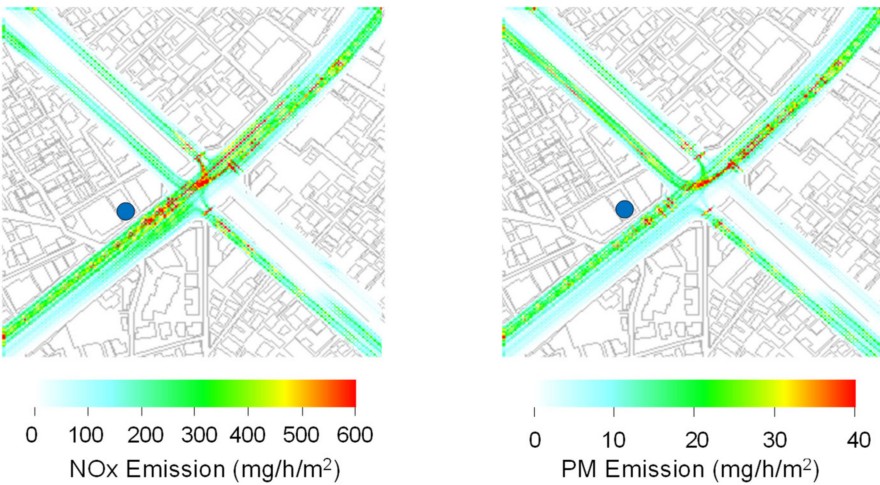

**Figure 18.** Estimated automobile emission distribution around the Kamiuma intersection (the blue circle indicates the Kamiuma Roadside Air Quality Monitoring Station (RAQMS)).

### 3.3. Analysis of the Flow Field by Computational Fluid Dynamics

To calculate the detailed airflow field by computational fluid dynamics (CFD), around a roadside, input data such as detailed buildings, elevated roads, and ground shapes of 1km square were created using 3D digital map data, and each mesh was created as an unstructured grid. Figure 19 shows an example of a calculation mesh around the Kamiuma intersection. The minimum mesh size was 0.75 m, and the total number of calculated cells was about 1.5 million cells. The wind field was calculated using the standard k-ε model as the turbulence model. Figure S10-3 shows the verification results by tracer gas analysis.

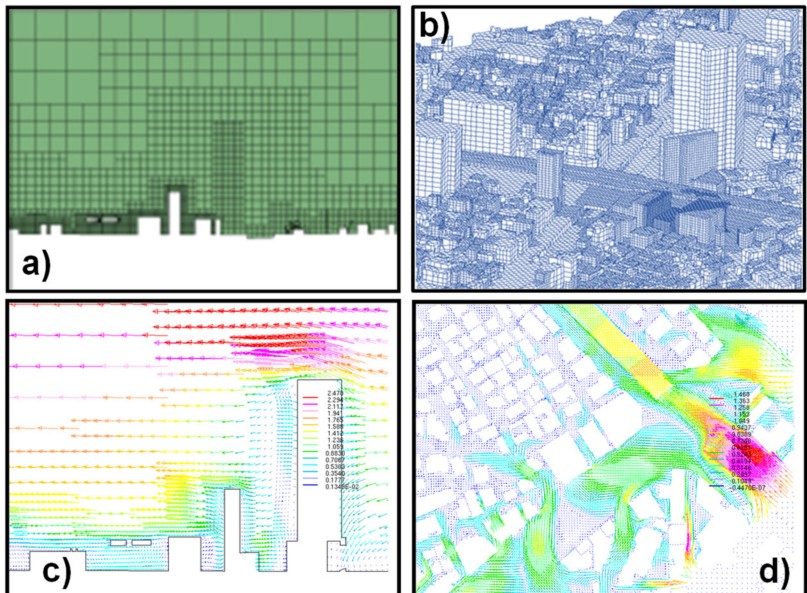

**Figure 19.** Roadside Air Quality Model structure and wind flow. (**a**) Cross section of calculation mesh. Fine mesh for central area and coarse mesh for surroundings. (**b**) Bird's-eye view of calculation mesh. (**c**) Calculated wind velocity field for cross section. (**d**) Calculated wind velocity field section approx. 2 m above the ground.

The automobile emission is discharged on the road with a 10 m mesh. It is assumed that the emissions discharged on the road are instantly and uniformly diluted in the grid. Diluted emissions are diffused by airflow over the model domain.

### 3.4. Roadside NO₂ Formation Process

Figure 20 shows the roadside NO$_2$ estimation process. Roadside NO$_2$ is estimated by dividing it into three types: background NO$_2$, primary NO$_2$ from vehicles, and secondary NO$_2$ due to ozone reaction in the roadside.

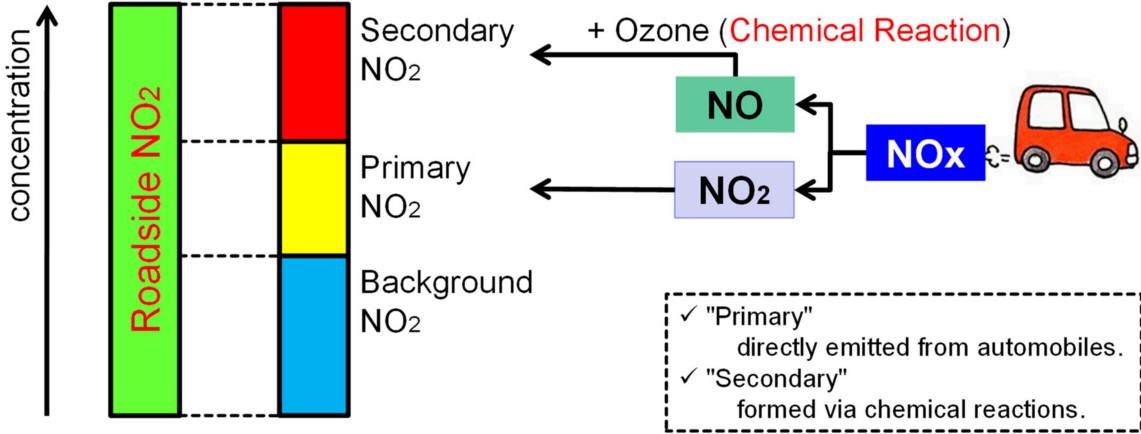

**Figure 20.** Roadside NO$_2$ estimation process.

Five NO$_2$ formation reactions with high reaction rates were incorporated into the Roadside Air Quality Model. Table 1 shows those reactions.

**Table 1.** NOx- O₃ chemical reactions and their rate coefficient.

| Reaction | Rate Constant | Unit |
|---|---|---|
| (R1) NO$_2$ + hν→ NO + O | $8.8 \times 10^{-3}$ | s$^{-1}$ |
| (R2) NO + O$_3$ → NO$_2$ + O$_2$ | $2.0 \times 10^{-12}$ exp $(-1400/T)$ | cm$^3$ molecule$^{-1}$ s$^{-1}$ |
| (R3) NO$_2$ + O → NO + O$_2$ | $6.5 \times 10^{-12}$ exp $(120/T)$ | cm$^3$ molecule$^{-1}$ s$^{-1}$ |
| (R4) O$_3$ + hν→ O + O$_2$ | $4.8 \times 10^{-4}$ | s$^{-1}$ |
| (R5) O + O$_2$ + M → O$_3$ + M | $6.0 \times 10^{-34}$ exp $(T/300)^{-2.3}$ | cm$^6$ molecule$^{-2}$ s$^{-1}$ |

### 3.5. Simulation Results

In 2008, JATOP conducted 2020 future air quality predictions to evaluate the effect of new regulations on diesel vehicles. The reason for the future prediction was that the replacement of new cars would be progressing, and the effect of the regulation could be confirmed. In this simulation, the Urban Air Quality Model was set with the same scenario to provide the boundary conditions for the Roadside Air Quality Model.

Three cases were calculated as a case study. Case A is the case where automobiles are naturally substituted under the conditions of exhaust gas regulations decided in 2008, Case B is where the high emitters described in Section 2.2.1 (2) do not exist, and Case C is the case where new emission regulations were introduced in 2014.

The calculation months are June and November, when the NO$_2$ concentration is high at the roadside. In June, O$_3$ is generated by a local photochemical reaction near the source and reacts immediately with NO, resulting in a high concentration of NO$_2$. In November, pollutants accumulate under stable weather conditions, resulting in high concentrations of NO$_2$.

Figure 21 shows the results of a Roadside Air Quality Monitoring Station (RAQMS) at the Kamiuma intersection (see Figure 18). The results posted as observations use a method that divides NO$_2$ into background NO$_2$, primary NO$_2$, and secondary NO$_2$ from the observed values [32]. Although the NO$_2$ concentration is decreased significantly even with natural substitution (Case A), the abundance of high emitters is low after the

2000 regulation (Figure 7), so there is little effect in Case B. With the introduction of new regulations in Case C, the $NO_2$ concentration is clearly lower than in Case A, but the reduction is not significant.

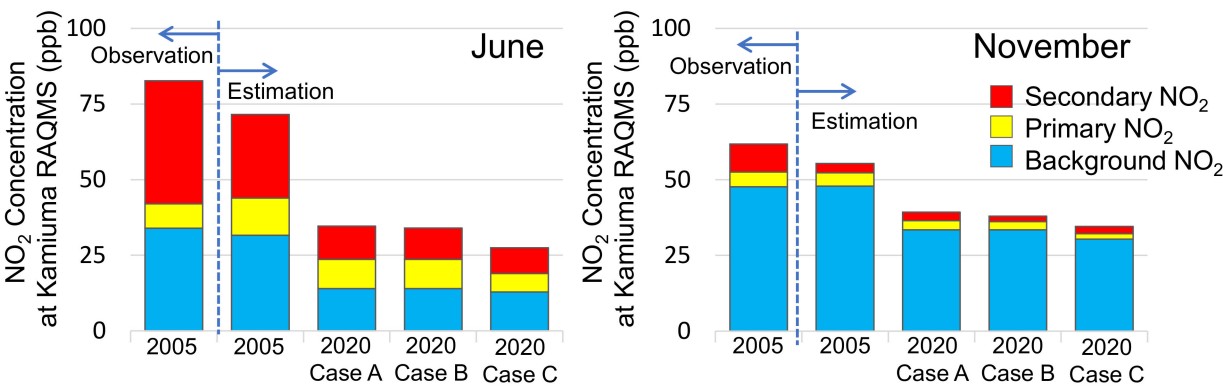

**Figure 21.** Simulation results of the Roadside Air Quality Model at the Kamiuma RAQMS.

Unfortunately, the Kamiuma RAQMS was suspended in December 2016, and it is unclear how accurately this estimation result predicted levels for 2020. Although, for reference, $NO_2$ concentration transitions of five RAQMS including Kamiuma, which are the top five highest $NO_2$ concentrations in Tokyo, are presented in S11 along with predicted results.

## 4. Active Use of Air Quality Study Results

### 4.1. Contribution to Policy Making Processes

During the 21 years of JCAP/JATOP Air Quality Model research, as shown in Figure 22, the research results, such as verification results of effects of novel emission control technology on the atmospheric environment, were reported to an expert committee on motor vehicle exhaust emission: the Central Environment Council of the Ministry of the Environment. Through these activities, JCAP/JATOP Air Quality Model research has contributed to policy discussions by providing scientific knowledge.

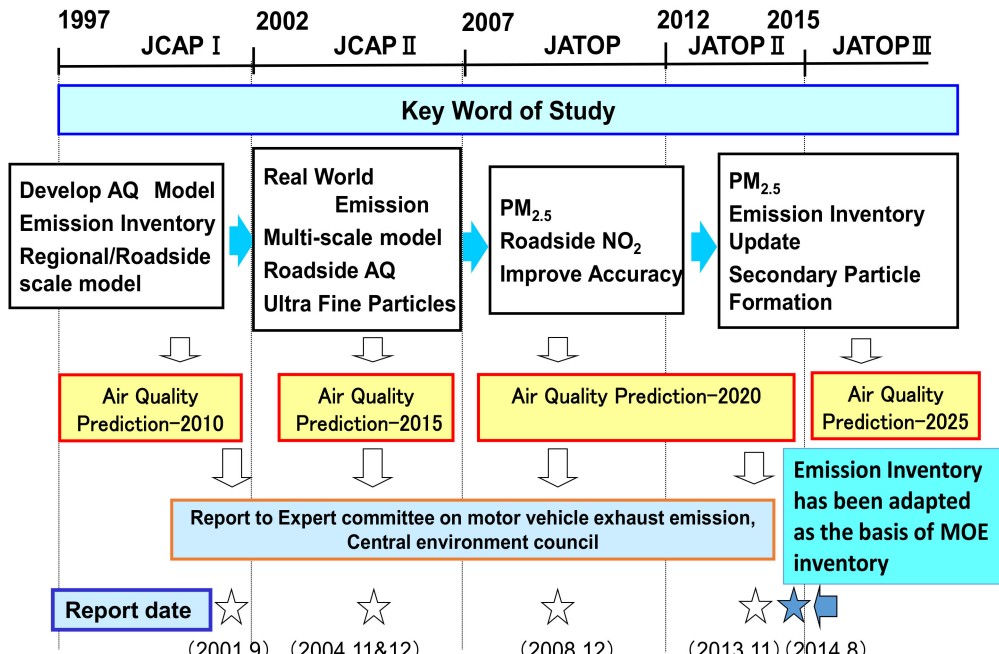

**Figure 22.** Contribution to policy making processes.

JCAP/JATOP Air Quality Model research also enabled the development of the Emission Inventory Model which is used as the basis for the national inventory.

### 4.2. JATOP Emission Inventory, JEI-DB Release

The emission inventory developed by JCAP/JATOP Urban Air Quality Model research (2000/2005/2010/2012) was released as JEI-DB. The PM and VOC profiles required to calculate chemical reactions are also partially dependent on SPECIATE [33] but were created using Japanese data wherever possible (Figure S12-1–3). Thus, JEI-DB is used for many Japanese atmospheric studies [34–37]. In addition, JEI-DB became the basis for air pollutant emission inventory "PM2.5 Emission Inventory (PM2.5EI)" of the Ministry of the Environment, and the inventories for 2012/2015 were created [38,39]. Currently, PM2.5EI for 2018 is being created. Figure 23 shows the transition of emissions in Japan since 2000, by JEI-DB and PM2.5EI.

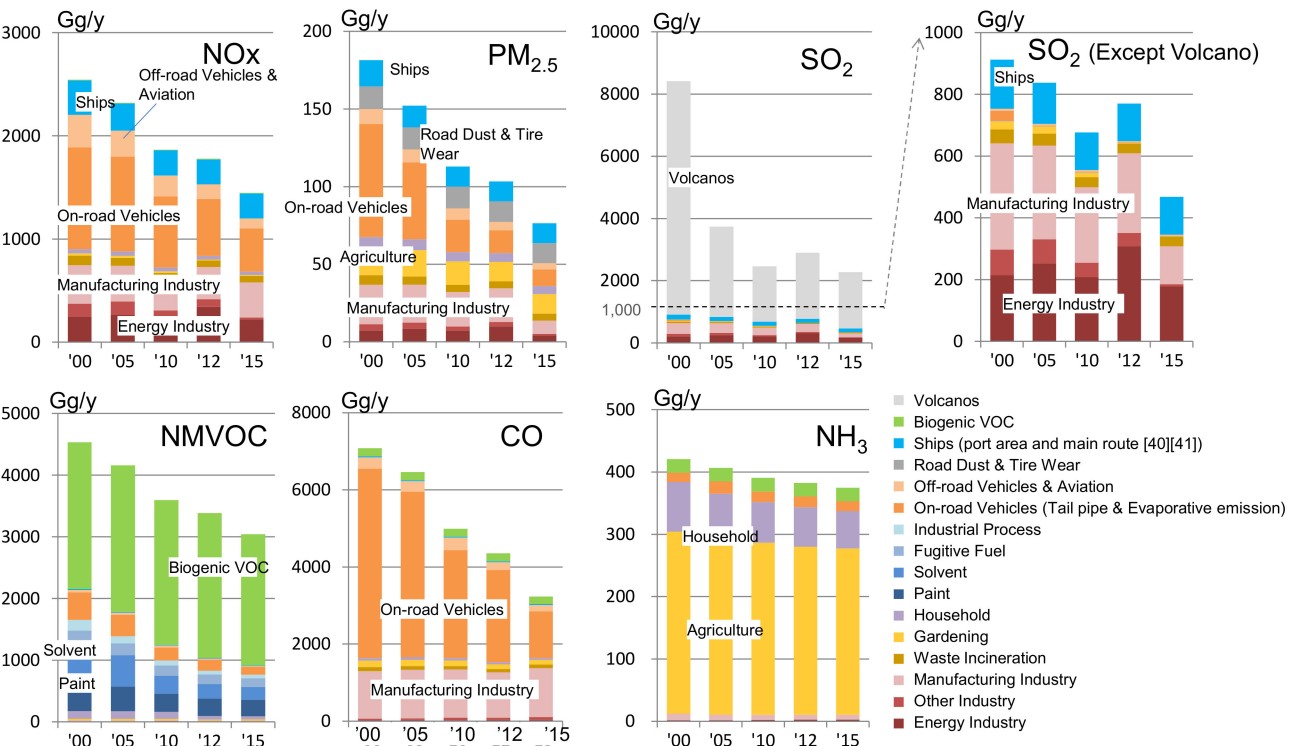

**Figure 23.** The transition of air pollutant emissions in Japan. All data are c.a. $1 \times 1$ km to c.a. $10 \times 10$ km nationwide, hourly by month. Ship emission was added to give an overview of total domestic emissions. Ship emission is limited to ports and major domestic routes [40,41].

JEI-DB provided 2000–2012 emission data from Japan's mobile sources for Phase 3 of the Global Emissions Inventory Grid Map produced by HTAP (the HTAPv3 mosaic inventory), an organization within the United Nations Economic Commission for Europe.

### 5. Conclusions

In JCAP/JATOP Air Quality Model research, by predicting future air quality using an atmospheric model, evaluating air quality improvement measures can provide effective scientific knowledge for air quality improvement policy discussions. In addition, our study is also able to contribute to the promotion of atmospheric research in Japan by disclosing the developed emission inventory.

The results of 21 years of JCAP/JATOP Air Quality Model research are:

1. Japanese emission inventory was created and provided the basis for the national inventory.

2.  The Regional Air Quality Model was applied to Japan to evaluate the effects of various measures and has provided scientific knowledge that continues to contribute to national environment policies.
3.  Disclosing the Air Quality Model and inventory contributed to the promotion of atmospheric research in Japan.

In addition, the future direction of Japan's atmospheric environment research was proposed, based on the research results of those 21 years.

The remaining issues are:

1.  Regarding emission inventory, the remaining issues are the maintenance of the national inventory/establishment of emission measurement methods from fixed sources/source profiles based on actual measurement data/improvement of spatiotemporal allocation methods/enhanced activity statistics/establishing an inventory maintenance system/updating system/ensuring consistency with other inventories.
2.  Regarding the Regional Air Quality Model, the remaining issues are improvement of reproducibility of the $PM_{2.5}$ component/elucidation, modeling of the SOA generation mechanism/elucidation, modeling of condensation/volatilization mechanism of condensable particles/elucidation, and modeling of the reaction mechanism of VOC component research.
3.  Development of evaluation methods such as future estimation and the elucidation of source contribution are still necessary.

**Supplementary Materials:** The following are available online at https://www.mdpi.com/article/10.3390/atmos12080943/s1, Figure S1: Suspended particle matter (SPM) as Japanese air quality regulation, S2: Vehicle category and emission regulation in Japan, S3: 103 Vehicle category for JEI-VEM, S4: Remote sensing device (RSD) measurement, S5: Ship emission, S6: Secondary organic aerosol (SOA) model results, S7: Reproducibility for the JATOP III Urban Air Quality Model, S8: Transient vehicle emission estimation model for the Roadside Air Quality Model, S9: Micro-scale Traffic Flow Model, S10: Tracer gas analysis for the Roadside Air Quality Model (Yoshikawa, 2003a), S11: Changes in monthly average concentration of $NO_2$ at the five top Roadside Air.Quality Monitoring Stations in Tokyo, S12: PM and VOC profiles from anthropogenic emissions. References [15,33,42–47] are cited in the Supplementary Materials.

**Author Contributions:** Writing—original draft preparation, Y.S.; writing—review and editing, T.M. All authors have read and agreed to the published version of the manuscript.

**Funding:** This study was conducted under research activities of the Japan Petroleum Energy Center (JPEC) by receiving a subsidy from the Ministry of Economy, Trade and Industry, Japan. This research received no external funding.

**Institutional Review Board Statement:** Not applicable.

**Informed Consent Statement:** Not applicable.

**Data Availability Statement:** Not applicable.

**Acknowledgments:** The authors would like to thank everyone involved in JCAP/JATOP Air Quality Model research. The following researchers participated in the atmospheric research group: Yamazaki, S.; Kobayashi, S.; Kunimi, H.; Kinugasa, Y.; Doki, S.; Yoshimura, M.; Nagao, M.; Wakita, M.; Matsunaga, S.; Ito, A.; Hagino, H.; Hayashi, S.; Hirai, H.; Minoura, H.; Chatani, S.; Takekawa, H.; Karasawa, M.; Terada, S.; Tanahashi, I.; Yoshikawa, Y.; Takada, T.; Misumi, A.; Fueki, A.; Uchida, K.; Nakatsuka, S.; Ashizaki, M.; Uchiyama, T.; Shimo, N. In addition, the Air Quality Model Expert Committee members provided advice from a professional standpoint: Wakamatsu, S.; Sakamoto, K.; Uno, I.; Ohara, T.; Kashima, S.; Kondo, H.; Takami, A.

**Conflicts of Interest:** The authors declare no conflict of interest.

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
