# Peer review of "Review of the JCAP/JATOP Air Quality Model Study in Japan"

_atmosphere, doi:10.3390/atmos12080943_

Round 1
Reviewer 1 Report
Dear authors,
Your manuscript presents an exhaustive and detailed revision of the JCAP/JATOP programs implemented from late 1990s in Japan to improve the air quality. The topic is of potential interest for the journal and the document is well organised and presented. I just have some minor comments before it can be accepted for publication at Atmosphere.
- It is not described in detail how your document may contribute to expand the existing knowledge or support the existing literature. In particular, you should be clearer and expand the aims of your revision.
- several very short sentences could be merged with others to improve the paper readability.
- There are too many figures but some of them could be moved to supplementary material.
- Several references are missed in the document.
- In conclusions, I suggest starting with a brief description of how your document may impact the air quality policies and programs around the world before describing the main findings. Including some future work would be desirable also.
Author Response
- I have changed "Introduction" to add how the obtained results have been utilized “Contribution to Policy Making Processes” and “Emission Inventory Release”
- Figure 7 and 14 have moved to Supplement
- Add reference 13, 21 and 22
- In Conclusion, I have changed to add how the obtained results have been utilized “Contribution to Policy Making Processes” and “Emission Inventory Release”

Reviewer 2 Report
The review is in the attached file

Author Response
I have changed Figure number link.

Reviewer 3 Report
It is very well constructed study. Therefore It can be accepted for publication without making any technical changes.
Author Response
Thank for your review.